# Decreased bioefficacy of long-lasting insecticidal nets and the resurgence of malaria in Papua New Guinea

Rebecca Vinit[1,8], Lincoln Timinao[1,2,8], Nakei Bubun[1,8], Michelle Katusele [1,8], Leanne J. Robinson[1,3], Peter Kaman[1], Muker Sakur[1], Leo Makita[4], Lisa Reimer [5], Louis Schofield[2], William Pomat[1], Ivo Mueller [6], Moses Laman [1], Tim Freeman[7] & Stephan Karl [1,2✉]

Papua New Guinea (PNG) has the highest malaria transmission outside of Africa. Long-lasting insecticidal nets (LLINs) are believed to have helped to reduce average malaria prevalence in PNG from 16% in 2008 to 1% in 2014. Since 2015 malaria in PNG has resurged significantly. Here, we present observations documenting decreased bioefficacy of unused LLINs with manufacturing dates between 2013 and 2019 collected from villages and LLIN distributors in PNG. Specifically, we show that of $n = 167$ tested LLINs manufactured after 2013, only 17% are fulfilling the required World Health Organisation bioefficacy standards of ≥ 80% 24 h mortality or ≥ 95% 60 min knockdown in bioassays with pyrethroid susceptible *Anopheles farauti* mosquitoes. In contrast, all (100%, $n = 25$) LLINs with manufacturing dates prior to 2013 are meeting these bioefficacy standards. These results suggest that decreased bioefficacy of LLINs is contributing to the malaria resurgence in PNG and increased scrutiny of LLIN quality is warranted.

[1] Vector-borne Diseases Unit, PNG Institute of Medical Research, PO Box 378, Madang, 511 Madang Province, Papua New Guinea. [2] Australian Institute of Tropical Health and Medicine, James Cook University, 1/14-88 McGregor Road, Smithfield, QLD 4870, Australia. [3] Burnet Institute, 85 Commercial Rd, Melbourne, VIC 3004, Australia. [4] National Department of Health, PO Box 807, Waigani Drive, Port Moresby, 131 NCD, Papua New Guinea. [5] Liverpool School of Tropical Medicine, Pembroke Pl, Liverpool, L3 5QA, UK. [6] Walter and Eliza Hall Institute of Medical Research, 1G, Royal Parade, Parkville, VIC 3052, Australia. [7] Rotarians Against Malaria, PO Box 3686, Port Moresby, 121 NCD, Papua New Guinea. [8] These authors contributed equally: Rebecca Vinit, Lincoln Timinao, Nakei Bubun, Michelle Katusele. ✉email: stephan.karl@jcu.edu.au

Papua New Guinea accounts for over 80% of malaria cases in the World Health Organisation (WHO) Western Pacific Region[1]. Long-lasting insecticidal nets (LLINs) are an important vector-control tool in malaria endemic countries including Papua New Guinea (PNG). Since the start of the distribution campaigns close to 2 billion LLINs have been delivered worldwide[2]. The global distribution of LLINs has contributed significantly to saving an estimated 6.8 million lives between 2000 and 2015[3].

In PNG, distributions started in 2006 and 12.8 million LLINs were delivered to the country between 2010 and 2019, with about 10.2 million since 2013. LLINs are the only vector-control tool implemented by the national malaria control program on a nationwide level[1,4]. From 2006 and until the end of 2019, the LLINs distributed within PNG were exclusively deltamethrin-treated PermaNet® 2.0 (Vestergaard–Frandsen). Globally, PermaNet® 2.0 had the largest market share in the LLIN industry in 2014[5]. Rotary Against Malaria (RAM) PNG is currently managing LLIN distribution in PNG.

Initially, LLIN distribution in PNG coincided with a massive decrease in malaria disease burden and infection prevalence from 15.7% (2008/2009) to 4.8% (2010/2011) and <1% (2013/2014)[6–8]. However, malaria indicators are on an upsurge in PNG since 2015, with a reported 9-fold increase (from <1% to 7.1%) in prevalence between 2013/14 and 2016/17[9,10]. The cause of this resurgence is currently not well understood but is likely due to multiple factors acting together. While PNG has experienced prolonged stock outs of antimalarial drugs during 2016–2017, it seems unlikely that drug shortages alone will lead to massive resurgence of transmission if >80% of infections are asymptomatic i.e. remain mostly untreated[11,12]. Malaria indicators also continued to rise after shortages of drugs had finished. Vector behavioural patterns may influence LLIN impact. In general, PNG malaria vectors, including the main coastal vector *Anopheles farauti* (*An. farauti*) tend to be exophagic and, with few exceptions, exhibit opportunistic host preference[13,14]. This is believed to be detrimental to LLIN impact as most human-vector contact occurs outside the house and vectors can easily seek alternative hosts thereby avoiding contact with the LLINs. In addition, studies suggest behavioural adaptation of local malaria vectors to LLINs, such that biting now occurs earlier in the evening[15,16]. Continued electrification (e.g. with solar-powered lights) allows people to be active much longer into the night than previously, even in remote PNG communities, which may enhance human-vector contact further. While insecticide resistance will have a detrimental effect on LLIN bioefficacy[17,18] regular insecticide resistance monitoring activities show no signs of emerging pyrethroid resistance in the anopheline mosquito populations in PNG since the beginning of the LLIN distributions[19]. This stands in contrast to recently found high levels of pyrethroid resistance in *Aedes aegypti* populations in PNG[20].

Reduced bioefficacy of LLINs can also be a result of substandard manufacturing process and distributions of substandard LLINs have occurred before, e.g. in Rwanda (2015) and Solomon Islands (2014)[21,22]. LLINs in use for 3 years or less should exhibit 80% 24 h mortality or 95% 60 min knockdown in standardised WHO cone bioassays[23,24]. Studies on the bioefficacy of PermaNet® 2.0 in use between 2000 and 2009 in PNG conducted by Katusele et al.[25] indicated that these LLINs were still highly effective even after 5 years of use, killing close to 100% of mosquitoes in standardised WHO cone bioassays. Only a slight reduction in bioefficacy in LLINs in use for more than 7 years was observed in these previous studies[25]. Similar results for PermaNet® 2.0 obtained in an African setting at the same time indicate that LLINs of this brand were fulfilling and exceeding WHO requirements during that period[26].

In order to understand whether LLIN bioefficacy may be a contributing factor to malaria resurgence in PNG, we show WHO bioefficacy test results from $n = 192$ brand new PermaNet® 2.0 LLINs still in original and unopened packaging collected either prior to distribution (for years 2018–2019) or from PNG communities (for years 2007–2017). Alongside, we present data from $n = 40$ used PermaNet® 2.0 LLINs collected in PNG communities. Our results indicate that LLINs distributed in PNG between 2013 and 2019 did not meet WHO bioefficacy standards. At the same time, PNG experienced a substantial resurgence in malaria.

## Results

**Cone bioassays with new LLINs.** Unused LLINs normally exhibit 100% 24 h mortality to fully pyrethroid (deltamethrin) susceptible anopheline mosquitoes[25,26]. Most of the new LLINs manufactured between 2007 and 2012 ($n = 25$) exhibited 100% 24 h mortality in the cone bioassays (21/25, 84.2%). All LLINs in this group also exhibited either ≥80% 24 h mortality or ≥95%

**Table 1 Number of new and unused LLINs and number of mosquitoes tested per year of LLIN manufacture, and resulting mosquito knockdown and mortality.**

| Year | LLINs tested | Mosquitoes tested | Mosquitoes KD/dead[a] | % KD$_{60min}$ (95% CI)[b] | % M$_{24h}$ (95% CI)[c] |
|---|---|---|---|---|---|
| 2007 | 1 | 25 | 25/25 | **100.00** (86.28-100) | **100.00** (86.28-100) |
| 2008 | 3 | 75 | 69/74 | **92.00** (83.4-97.01) | **98.67** (92.13-100) |
| 2009 | 2 | 50 | 50/50 | **100.00** (92.89-100) | **100.00** (92.89-100) |
| 2010 | 7 | 175 | 173/173 | **98.86** (95.93-99.86) | **98.86** (95.93-99.86) |
| 2012 | 12 | 300 | 286/295 | **95.33** (92.29-97.43) | **98.33** (96.15-99.46) |
| Sub-total (2007–2012) | 25 | 625 | 603/617 | **96.48** (94.72-97.78) | **98.72** (97.49-99.45) |
| 2013 | 20 | 500 | 243/241 | **48.60** (44.14-53.08) | **48.20** (43.74-52.68) |
| 2014 | 9 | 225 | 68/65 | **30.22** (24.3-36.68) | **28.89** (23.06-35.29) |
| 2015 | 24 | 600 | 339/369 | **56.50** (52.43-60.51) | **61.50** (57.47-65.41) |
| 2016 | 11 | 275 | 90/55 | **32.73** (27.21-38.62) | **20.00** (15.44-25.22) |
| 2017 | 54 | 1350 | 395/338 | **29.26** (26.84-31.77) | **25.04** (22.75-27.44) |
| 2018 | 27 | 662 | 396/385 | **59.82** (55.97-63.58) | **58.16** (54.29-61.95) |
| 2019 | 22 | 550 | 185/217 | **33.64** (29.69-37.76) | **39.45** (35.35-43.68) |
| Sub-Total (2013–2019) | 167 | 4162 | 1716/1670 | **41.23** (39.74-42.73) | **40.12** (38.65-41.62) |
| Overall | 192 | 4787 | 2319/2287 | **48.44** (47-49.9) | **47.78** (46.4-49.2) |

Source data for this table are provided as a Source Data file.
[a]Number of mosquitoes knocked down (KD) after 60 min and number of mosquitoes dead after 24 h (dead).
[b]Mean percent 60 min knockdown (KD$_{60min}$, bold) with exact 95% confidence intervals of the proportions (95% CIs, in parentheses).
[c]Mean percent 24 h mortality (M$_{24h}$, bold) with exact 95% confidence intervals of the proportions (95% CIs, in parentheses).

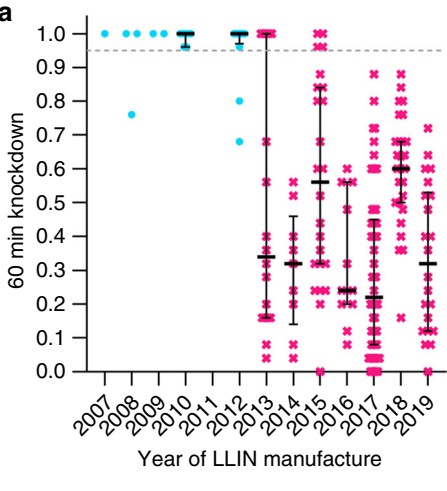

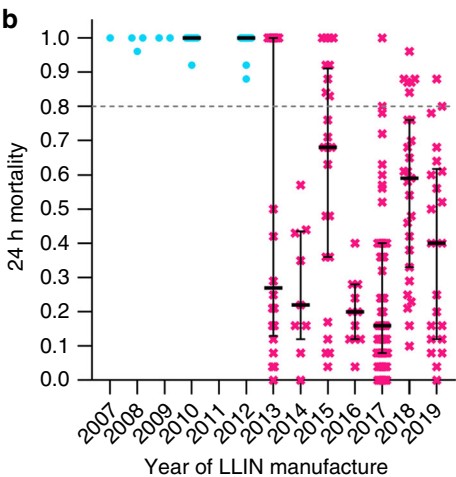

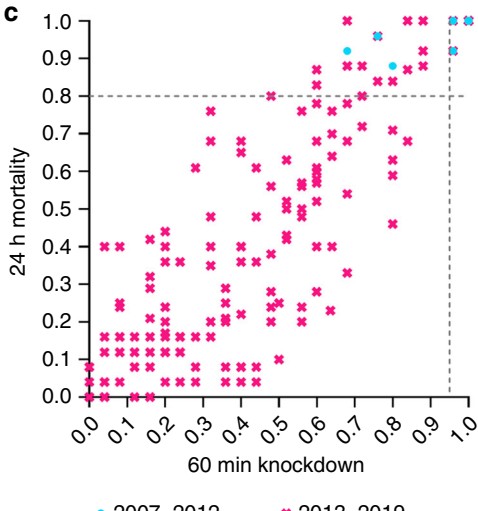

• 2007–2012   ✕ 2013–2019

**Fig. 1 WHO cone-bioassay data for new and unused LLINs by year of manufacture. a** 60 min knockdown rates of new LLINs. **b** Adjusted 24 h mortality rates of new LLINs. **c** Correlation between 60 min knockdown and adjusted 24 h mortality rates. The dashed lines indicate 95% 60 min knockdown (**a** and **c**) and 80% 24 h mortality (**b** and **c**) rates. The colours and symbols differentiate between 2007–2012 LLINs (turquois circles) and 2013–2019 LLINs (magenta crosses). Error bars in panels **a** and **b** are medians and interquartile ranges (only presented for sample sizes of $n \geq 7$). Each data point represents one bioassay result conducted on one individual, independent LLIN (number of LLINs tested per year given in Table 1 in the 'LLINs tested' column). If the bioassay was valid (negative control mortality <10%), bioassays on individual LLIN samples were not repeated. Source data for this Figure are provided as a Source Data file.

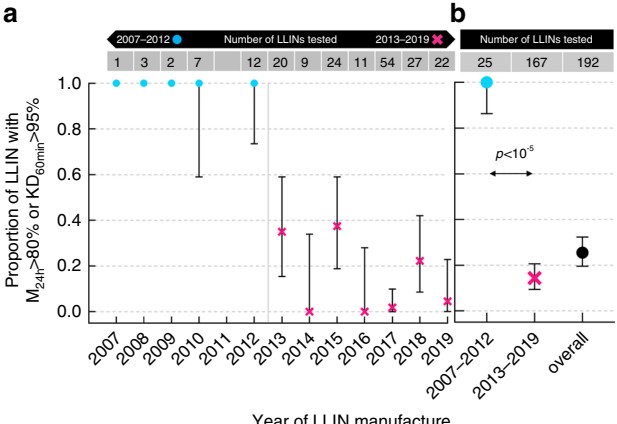

**Fig. 2 Proportion of new and unused LLINs with adequate bioefficacy by year of manufacture. a** Proportion of new and unused LLIN exhibiting ≥95% 60 min knockdown ($KD_{60min}$) or ≥80% adjusted 24 h mortality ($M_{24h}$) grouped by year of manufacture. **b** Proportion of LLINs exhibiting ≥95% 60 min knockdown ($KD_{60min}$) or ≥80% adjusted 24 h mortality ($M_{24h}$) for groups with years of manufacture between 2007–2012 (turquois circles) and from 2013 to 2019 (magenta crosses), as well as overall (2007–2019, black circle). The table on top of the panels indicates the numbers of individual LLINs tested in each year or group. Data are presented as mean proportions and their exact 95% confidence intervals. The p-value between the 2007–2012 and 2013–2019 groups is the result of a Chi-squared test (2-sided) comparing the proportions in these two groups. The exact z-score of the test is 8.442. Source data for this Figure are provided as a Source Data file.

Table 1 shows summary data for all new LLINs tested in the present study grouped by the year of manufacture.

Figure 1 shows 24 h mortality rates and 60 min knockdown rates versus year of LLIN manufacture, and correlation of these two measures for the $n = 192$ unused LLINs tested in this study.

Figure 2 shows the decline in the proportion of unused LLINs exhibiting either ≥80% 24 h mortality or ≥95% 60 min knockdown by year of manufacture. Data are also presented grouped into 2007–2012 and 2013 to 2019 groups.

**Cone bioassays with used LLINs.** Table 2 shows summary data for all used LLINs ($n = 40$) tested in the present study grouped by owner-reported LLIN usage duration of either more or less than 3 years.

The proportions of mosquitoes knocked down after 60 min in the two groups were 51% and 48%, respectively, and the

knockdown indicating that these nets would likely have passed a phase 3 bioefficacy study at baseline as per WHO guidelines[23].

In striking contrast, of the unused LLINs manufactured between 2013 and 2019 only 7% (12/167) exhibited 100% 24 h mortality and the proportion of nets exhibiting either ≥80% mortality or ≥95% knockdown reduced from 100% (95% confidence interval (CI) 86–100%) in the 2007–2012 group to 17% (95% CI 9–20%) in the 2013–2019 group. The difference was highly statistically significant (Chi-squared equal to 75.4; p-value < $10^{-5}$).

**Table 2 Summary of the number of LLINs and number of mosquitoes tested according to owner-reported usage duration.**

| Owner-reported usage duration | LLINs tested | Mosquitoes tested | Mosquitoes KD/dead[a] | % KD$_{60min}$ (95% CI)[b] | % M$_{24h}$ (95% CI)[c] |
|---|---|---|---|---|---|
| 1–3 years | 14 | 280 | 143/135 | **51.07** (45.05–57.07) | **48.21** (42.23–54.24) |
| >3 years | 26 | 531 | 278/335 | **52.35** (48.01–56.67) | **63.09** (58.83–67.20) |
| Overall | 40 | 811 | 421/470 | **51.91** (48.41–55.40) | **57.95** (54.47–61.38) |

Source data for this table are provided as a Source Data file.
[a]Number of mosquitoes knocked down (KD) after 60 min and number of mosquitoes dead after 24 h (dead).
[b]Mean percent 60 min knockdown (KD$_{60min}$, bold) with exact 95% confidence intervals of the proportions (95% CIs, in parentheses).
[c]Mean percent 24 h mortality (M$_{24h}$, bold) with exact 95% confidence intervals of the proportions (95% CIs, in parentheses).

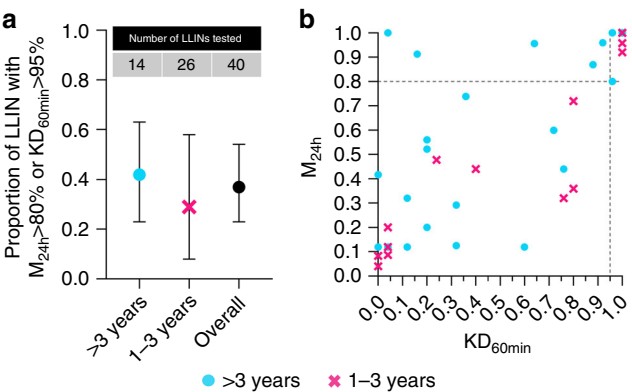

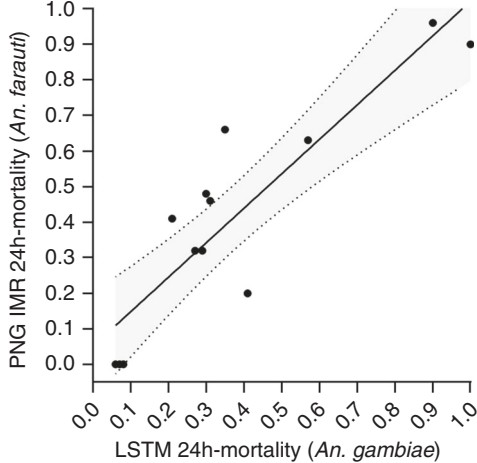

**Fig. 3 Bioefficacy of used LLINs. a** Owner-reported LLIN usage duration versus the proportion of LLINs with either ≥95% 60 min knockdown (KD$_{60min}$) or ≥80% 24 h mosquito mortality (M$_{24h}$). Data are presented as mean proportions and their exact 95% confidence intervals. Data from LLINs with years of manufacture between 2007–2012 are presented as turquois circles and from 2013 to 2019 as magenta crosses. Overall data (2007–2019) is presented as black circle. The table on top of the panels indicates the numbers of individual LLINs tested in each year or group. **b** KD$_{60min}$ versus adjusted M$_{24h}$ for all used LLINs tested ($n = 40$). The dashed lines indicate 95% KD$_{60min}$ and 80% M$_{24h}$ thresholds. Each data point represents one bioassay result conducted on one individual, independent LLIN. If the bioassay was valid (positive and negative control mortality <10%), bioassays on individual LLIN samples were not repeated. Source data for this Figure are provided as a Source Data file.

**Fig. 4 Correlation of cone-bioassay data (24 h mortality) for *An. farauti* and *An. gambiae* G3.** Experiments with *An. farauti* were conducted at PNGIMR and experiments with *An. gambiae* G3 were conducted at LSTM with using the same LLIN samples ($n = 19$). The best fit curve obtained using standard linear regression analysis is represented by the continuous black line. The grey shading in between the dotted lines represents the 95% confidence band of the best fit curve. Each data point represents one bioassay result conducted on one individual, independent LLIN. If the bioassay was valid (positive and negative control mortality <10%) bioassays on individual LLIN samples were not repeated. Source data for this Figure are provided as a Source Data file.

proportions of mosquitoes dead after 24 h in the two groups were 52% and 63%, respectively. The differences between the groups were not statistically significant.

At least 80% of used LLINs should exhibit 95% 60 min knockdown or 80% 24 h mortality if they are less than 3 years in use[23,24]. Figure 3 shows owner-reported duration of LLIN usage versus the proportion of LLINs meeting this condition for the $n = 40$ used LLINs tested in the present study.

Owner-reported LLIN usage duration was not significantly associated with observed LLIN bioefficacy and overall 37% (95% CI: 23–54%) of the LLINs met the performance criterion.

**Confirmatory bioassays**. Replicate results obtained with pyrethroid (deltamethrin) susceptible standard *An. gambiae* s.s. strain G3 corresponded well to the results obtained in PNG with *An. farauti* as shown in Fig. 4 (coefficient of determination equal to 0.80) indicating that results apply to susceptible Anopheles species in other (e.g. African) settings.

**Experiments simulating LLIN container storage**. Data on the temperature distribution in the storage container over the 5-day measurement period are presented in Supplementary Fig. 1. In

summary, temperatures away from the immediate inner ceiling surface of the container never exceeded 40 °C during the 5-day measurement period in the hot normal conditions of Port Moresby summer (February 2019). Only at the immediate inner surface of the top of the container temperatures over 50 °C were measured. However, even there, temperatures remained below 50 °C for 94% of the measurement period. In corresponding experiments simulating container storage, we did not observe any reduction in 24 h mortality or 60 min knockdown rate when LLINs with a confirmed 100% 24 h mortality (manufacture year 2012) were exposed to 6 weeks of storage at 60 °C.

**Discussion**

Our findings show that LLINs distributed in PNG between 2013 and 2019 have not been exhibiting the required bioefficacy prior to use, whereas LLINs distributed before 2013 performed significantly better[25]. This observation coincides with a substantial resurgence of malaria in many parts of the country[1,9,10].

Specifically, all LLINs tested in the present study that were manufactured between 2007 and 2012 ($n = 25$) exhibited either ≥80% 24 h mortality or ≥95% 60 min knockdown. In contrast, only 17% of LLINs ($n = 167$) that were manufactured between

2013 and 2019 met these criteria. However, all LLINs should meet these criteria.

These results span 78 separate LLIN batches. Results from 2013 indicate that some LLIN batches in that year were still performing as expected. Specifically, batch # 1 258 13 (3) from which $n = 5$ LLINs were tested, exhibited 100% knockdown and 100% mortality for all five of these LLINs. The other five LLIN batches tested from 2013 all exhibited diminished bioefficacy. After 2013, none of the batches from which multiple LLINs were available exhibited bioefficacy results meeting aforementioned criteria for all LLINs sampled from a specific batch. This indicates that the change occurred in 2013.

Although the number of LLIN tested for the 2007–2012 period is relatively small due to the difficulty of obtaining unused LLINs from such a long time ago, the difference observed between the 2007–2012 and the 2013–2019 groups is large and statistically significant (Chi-squared equal to 75.4; $p$-value $< 10^{-5}$; z-score 8.4).

We adhered strictly to WHO guidelines in terms of cone-bioassay setup and conditions. However, since this was a research study involving a registered and certified product, rather than a phase 3 trial aimed at obtaining WHO certification for a new product, less mosquitoes per individual LLIN were exposed. Specifically, in a phase 3 LLIN trial, usually five sections from 30 unused LLINs are tested with 20 mosquitoes each, resulting in 100 exposed mosquitoes per LLIN and a total of 3000 exposed mosquitoes per study[23]. Operational monitoring guidelines recommend exposure of 40 mosquitoes per LLIN and a total of 30 LLINs to be tested, resulting in a total of 1200 mosquitoes exposed[24]. In the present study, we aimed to test as many unused LLIN from previous years as possible resulting in bioefficacy tests on $n = 192$ LLINs from $n = 78$ batches across 12 years. This is much more than would normally be tested in a phase 3 LLIN trial at baseline. However, we tested only one section per LLIN using 25 mosquitoes, resulting in 4787 mosquitoes exposed to LLINs in cone bioassays. This study has the limitation that we did not account for within-LLIN variability in our analyses. However, we have sampled from a very broad range of LLINs, exposing a large number of susceptible mosquitoes, resulting in overwhelmingly statistically significant results. Thus, while testing multiple sections per LLIN in the precertification phase is important to ensure product quality, it is of less relevance to the conclusions of this study with regards to the diminished bioefficacy of a large number of unused LLINs. For further confirmation, we tested $n = 5$ sections (four sides + roof) from one net from each of the years 2007, 2008, 2009, 2010 and 2012 with 20 mosquitoes each. All 25 of these bioassays showed 100% 60 min knockdown and 100% 24 h mortality.

Since used LLINs may be subject to more wear on the side panels than on the roof, our methodology may have resulted in an overestimation of the used LLINs failing the ≥80% 24 h mortality or ≥95% 60 min knockdown criteria for the used LLIN category, which is a limitation of this study.

PermaNet® 2.0 has been among the most widely distributed LLIN brand in the recent decade and bioefficacy studies have been conducted in several countries. Most of these studies used LLINs dating back to 2012 and before, and showed very good bioefficacy of PermaNet® 2.0[27–31]. In tests conducted in an African setting (Northern Tanzania), PermaNet® 2.0 from 2017 were reported to fulfil all requirements[32]. However in a recent report from Iran, PermaNet® 2.0 did not seem to fulfil bioefficacy requirements[33]. Studies also investigated the effect of developing pyrethroid resistance on PermaNet® 2.0 bioefficacy, e.g. refs. [17,34]. Unsurprisingly, pyrethroid resistance reduced the knockdown and mortality rates of the exposed mosquitoes.

While storage at elevated temperatures could potentially be detrimental to LLIN bioefficacy (we are not aware of published evidence for this), we consider it unlikely that this is responsible for the diminished bioefficacy of the LLINs as observed in the present study. Firstly, the $n = 25$ LLINs manufactured between 2007 and 2012 (obtained from several different provinces and 19 different production batches) tested in this study had been stored in tropical climate for up to 12 years before testing. All 25 of these LLINs exhibited either ≥80% 24 h mortality or ≥95% 60 min knockdown. Secondly, all samples of unused LLINs in their original and unopened packaging from 2018 and 2019 were taken from the coolest, most central parts of the containers where temperatures do not appear to exceed 40 °C (temperature data from an LLIN container in Port Moresby can be found in Supplementary Fig. 1). Thirdly, our experiments where LLINs from 2012 (confirmed 100% 24 h mortality) were exposed to 6 weeks storage at 60 °C showed no signs of reduced 24 h mortality or 60 min knockdown rate when tested with susceptible An. farauti mosquitoes.

It has been noted that the manufacturer made structural changes to the LLINs over the years. Bails of 100 LLINs weighing 50 kg in 2012 weighed as little as 43 kg in 2019, which translates to a weight reduction of 70 g per net. While these changes may be explained in terms of changing knitting weave, it calls into question if other changes have taken place and whether these LLINs have consequently been tested again to confirm bioefficacy (information on average weight of LLIN bails vs year of manufacture can be found in Supplementary Fig. 2).

All LLINs distributed in PNG undergo quality assurance procedures that include chemical insecticide content validation and, in recent years, wash tests. These procedures are conducted by Crown Agents and TÜV SÜD. Insecticide content and wash index for the LLIN consignments in the present study were reported to be within the prescribed ranges by these agencies throughout the period of 2007–2019. A hypothesis arising from this is that, while the LLINs may have the full concentration of insecticide, the availability of insecticide may be restricted on the surface of the LLINs. However, it appears that these predelivery inspections provide little information about bioefficacy. Therefore, bioefficacy testing should be included in the predelivery inspections.

Our findings suggest that only a small proportion of PermaNet® 2.0 LLINs distributed in PNG in recent years met quality standards with respect to bioefficacy, and that LLINs with reduced and highly variable bioefficacy were distributed since at least 2013. This may have seriously affected malaria control efforts in the country, and it is not unlikely that other countries may have been affected as well.

The most recent malaria indicator survey in PNG dates back to 2016 and 2017 and has reported a massive increase in malaria prevalence in PNG from <1% to 7.1%[10]. These results are corroborated by a rising number of clinical cases reported from many parts of the country[1]. While it is not possible to unequivocally attribute the coincidental rise in malaria cases over the same period in PNG to the observations presented in this study, we consider it highly likely that reduced bioefficacy of the LLINs is at least in part responsible for this malaria resurgence (likely in addition to the drug shortages and behavioural changes in mosquitoes and humans described above). An additional negative effect of exposure to sublethal concentrations of insecticide on the LLINs is that the chances for the emergence of insecticide resistance may be increased[35].

Few countries seem to conduct regular quality control of LLINs received for distribution using cone bioassays[36]. Our present study suggests that it may be of benefit to recipient countries to implement this type of quality assessment to prevent distribution

of LLINs with compromised bioefficacy and thus reduced utility for malaria control. To investigate whether the observed problem is restricted to PNG or if other countries and LLIN products are also affected, we recommend that quality assurance through operational monitoring of LLINs in all malaria endemic countries is conducted and capacity of endemic countries to conduct the required studies is strengthened.

Due to the widespread pyrethroid resistance in many countries, access to insectary facilities and susceptible mosquito strains is required to perform these tests. Further studies are required to determine the underlying cause of the observed reduced bioefficacy of these LLINs.

## Methods

**Origin of tested LLINs.** Unused LLINs manufactured in 2018 and 2019 ($n = 49$) were provided by RAM PNG from consignments dedicated to different PNG provinces, whereas unused LLINs manufactured in 2007–2017 ($n = 143$) were obtained from villages or provincial health authorities in various PNG provinces. All LLINs were still in original and unopened packaging,

Overall, $n = 192$ unused LLINs with 78 different batch numbers were tested. These had been distributed to or were intended for distribution in 15 PNG Provinces, namely Central, Chimbu, East New Britain, East Sepik, Eastern Highlands, Gulf, Hela, Manus, Morobe, New Ireland, Oro, Southern Highlands, Western, Western Highlands and West New Britain (details are provided in Supplementary Table 1).

Used LLINs ($n = 40$) were collected in communities in Madang Province and Gulf Province in 2018 and 2019, and owners were asked to indicate how long they had been using these LLINs. These LLINs showed signs of wear and were not in original packaging.

**Temperature in storage shipping containers containing LLINs.** Temperature was logged over a period of 5 days in four locations in a shipping container filled with LLIN in Port Moresby. Temperature loggers (USB Data Logger RC5, Elitech) were placed into four locations inside the container (i) immediately beneath the ceiling; (ii) in the centre of the topmost layer of LLIN bails; (iii) immediately beneath the topmost layer of LLIN bails; (iv) in the centre of the container.

To simulate container storage at elevated temperatures, $n = 3$ LLINs with confirmed 100% bioefficacy (manufacture date 2012) were exposed to a temperature of 60 °C in an oven for 6 weeks and bioefficacy was tested weekly.

**Origin of exposed mosquitoes.** Mosquitoes used in this study were either drawn from a mosquito colony established at PNGIMR or collected locally as larvae and reared to adult stage. Specifically, the laboratory colony used in the present study was an *An. farauti* strain originally from Rabaul, East New Britain province, PNG. This colony has existed for many decades and was back-crossed with local *An. farauti* mosquitoes from Madang province in PNG several times, the last time in 2012. Regular WHO-tube testing on the colony mosquitoes before, throughout and after completion of this study confirmed susceptibility to deltamethrin, lambda-cyhalothrin, dichlorodiphenyltrichloroethane (DDT), bendiocarb and malathion insecticides. Local wild-caught *An. farauti* mosquitoes used in this study originated from larval collections along the North Coast of Madang Province, PNG. Extensive insecticide resistance monitoring using WHO-tube assays was conducted prior to, during and after this study and with larvae collected from the same habitats, confirming full susceptibility of these wild-caught mosquitoes to deltamethrin, lambda-cyhalothrin, DDT, Bendiocarb and Malathion insecticides. Overall, there is no indication of pyrethroid, and especially deltamethrin, resistance in Anopheline populations from anywhere in PNG[19].

**WHO cone bioassays.** WHO cone bioassays were conducted on the LLINs according to WHO guidelines[23,24], using 25 (as recommended in ref. [37]) fully pyrethroid susceptible *An. farauti* mosquitoes.

The experimental setup of the WHO cone bioassays is shown in Supplementary Fig. 3.

Mosquitoes were 3–5-days-old when used in the cone bioassays. Tests were conducted in ambient tropical environment (Madang, PNG, latitude 5° South), and temperature and humidity requirements were met in all assays included in the study. The number of mosquitoes per cone was $n = 5$ and exposure time to the LLINs was 3 min.

All cone bioassays included positive and negative controls. We used LLINs manufactured in 2012 and with a known 100% 24 h mortality as positive controls and pieces of untreated netting as negative controls. Results were excluded if 24 h mortality in the negative control exceeded 10%. Test results were adjusted using 'Abbott's formula' when negative control 24 h mortality was >0% and ≤10%.

After exposure to the LLINs, mosquitoes were gently transferred from the cones to cardboard holding cups screened with untreated netting and provided access to 10% sugar solution via a soaked piece of cotton wool placed on top of

the netting. After 60 min, the number of mosquitoes knocked down in the holding cups was enumerated and after 24 h, the number of dead mosquitoes in the holding cups was enumerated.

The total number of mosquitoes exposed to each of the 192 unused LLIN sections was $n = 25$ in five cones and the number of mosquitoes exposed to each of the 40 used LLIN sections was $n = 20$ in four cones. One LLIN side panel section ($30 \times 30$ cm) was tested per LLIN as shown in Supplementary Fig. 3, placing the cones onto different parts of the section. This approach used a lower number of mosquitoes per LLIN as recommended in the WHO guidelines for large scale (phase 3) LLIN candidate product evaluation where $n = 5$ sections from $n = 30$ unused LLINs are tested with $n = 20$ mosquitoes each (i.e. $n = 100$ mosquitoes per LLIN) and resembled more closely the approaches recommended for testing LLINs under operational conditions[24] The reason we chose this different approach is that we aimed to test a much larger number of LLIN samples from an already WHO certified product to span as many years and batches as possible.

Of the LLINs tested, $n = 19$ (including one positive and one negative control) were sent to Liverpool School of Tropical Medicine (LSTM) for confirmation of results through blinded repetition of the WHO cone bioassays with a very similar Standard Operating Procedure based on WHO guidelines and the *An. gambiae* G3 standard pyrethroid susceptible laboratory strain. The LSTM laboratory used $n = 50$ mosquitoes (10 cones) per LLIN sample to conduct the repetitions.

**Data analysis and statistics.** Data were collected using the Epicollect 5 (Imperial College, London) open access electronic data capture system. Data were analysed using Microsoft Excel 2016 (Microsoft Inc.) and GraphPad Prism 8.0 (GraphPad Software). No custom computer code was used.

The main outcome variables of WHO cone bioassays are 24 h mosquito mortality and 60 min mosquito knockdown. Specifically, an LLIN should exhibit either ≥80% 24 h mosquito mortality or ≥95% 60 min mosquito knockdown. These are proportions and as such, wherever appropriate, exact 95% confidence intervals of proportions (Clopper–Pearson) were used to quantify the uncertainty in a proportion estimate due to sample size[38]. To summarize proportion data (e.g. Fig. 1), medians and interquartile ranges are used. For correlation data (e.g. Fig. 4) we used parametric correlation statistics (Pearson) and linear regression, also depicting the 95% confidence bands of the best fit curve. This is appropriate since even though the data presented originate from inherently non-normal distributions they are continuous, paired and exhibit equal variances.

**Reporting summary.** Further information on research design is available in the Nature Research Reporting Summary linked to this article.

## Data availability

Figures 1, 3 and 4 contain raw data. All data collected in this study are available in the form of an excel spreadsheet provided as Supplementary Data 1 or under https://doi.org/10.6084/m9.figshare.12552137.v1. Source data are provided with this paper.

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

## Acknowledgements

We would like to thank all PNG communities who provided LLINs for testing and the RAM teams for collecting the LLINs. This study was funded in part by the Global Fund to Fight Aids, Tuberculosis and Malaria. L.T. is supported by a PhD Scholarship from James Cook University. M.K. is the recipient of a Wellcome Trust International Masters Fellowship. S.K. (GNT1141441) and L.J.R. (GNT1161627) are recipients of an Australian National Health and Medical Research Council (NHMRC) Career Development Fellowship.

## Author contributions

R.V., L.T., N.B. and M.K. conducted all field and laboratory work with assistance from P.K. and M.S. L.J.R., I.M., W.P., L.M., L.S. and M.L. provided critical review of the data and manuscript. L.R. conducted the confirmatory bioassays with *An. gambiae*. T.F. and S.K. conceived the study, analysed the data and wrote the first manuscript draft.

## Competing interests

The authors declare no competing interests.

## Additional information

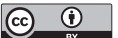

