## [Peer Review File · Nature Communications]

In their review of the first version of this manuscript, reviewer 2 added their comments to the manuscript file. These comments, excluding minor textual revisions, have been copied into this Peer Review File.

Review: Decreased bioefficacy of long-lasting insecticidal nets and the resurgence of malaria in Papua New Guinea

The paper presents data that support the authors' premise that quality of Insecticide Treated Nets (ITNs) distributed after 2012 is substantially lower than those distributed before this date. This is the first data of its kind that I have seen although rumours of reduced quality of ITNs

This is an extremely important paper of wide interest to the malaria control community globally. I feel it is critical that the issue of net quality is highlighted and its possible impact on malaria control is brought to the attention of the wider community. It is well known among many that net manufacturers cut corners to reduce costs and quality of nets. Since the cost of a net has fallen by around \$5 over the past 10 years it is inevitable that companies are under pressure to make cheaper nets. This may result in lower fabric weight and / or reduced insecticidal load which will impact bioefficacy even among susceptible mosquitoes. This appears to have been observed in the data reported here. Much emphasis is put on the importance of insecticide resistance in reducing net efficacy by the malaria control community. Here we are provided with good data from a country without insecticide resistance that suggest that manufacturing processes may be compromising net bioefficacy.

The data provided is unique because it is extremely difficult to find old unused nets in the field and the design is novel. The authors ensured that their findings were independently validated in a second laboratory which adds weight to the conclusions. The discussion could be further developed and the discussion broadened.

The writing is clear and concise and the claims are reasonable. More review in the introduction and discussion would benefit the paper. The analysis should be improved and the methods more clearly written. There are no ethical issues.

The paper is of sufficient quality to be published provided the authors make a number of compulsory revisions. If these are met then the paper should be resubmitted.

- 1) The malaria control community uses WHO guidance to test the quality of Insecticide treated nets. The authors need to revise their findings so that the results following WHO standard reporting are included i.e. the proportion of nets that pass or fail WHO bioefficacy criteria by year. This is important to allow comparison with other studies and to ensure that the reporting meets recognised standards.
- 2) Authors should provide a table of results in addition to the multiple figures included in the manuscript. The table should include the number of nets tested by year, the number of mosquitoes tested % (95% CI) knock down % (95% CI) 24 hour mortality.
- 3) I am concerned that 2007 to 2010 have very small sample sizes. I would be happier if the nets from 2007 to 2012 were combined to one category for analysis and analysis was conducted on samples from before and after 2012.

- 4) I do however like very much the data presentation that the authors have used, especially in figure 2 and the additional analysis should be included just to ensure that conventional reporting is also included for completeness and improved acceptability to all members of the malaria community, in particular manufacturers who should not be left with any grounds to contest the results.

- 5) The authors need to spend some time editing the methods section as it is not possible to follow exactly what was done as written. I have provided more detailed suggestions in the tables. Once this is done it may be possible to evaluate whether the correct WHO procedures were performed. If they were not followed then the results must be justified to ensure that sufficient replication per net was used.

- 6) The study has a number of limitations such as the small sample size and the unknown storage conditions of the unopened nets found in houses. This should be addressed in the discussion.

Major

Line	Phrase	Comment
Intro 46-48	Bioefficacy of LLINs in PNG appears to have been highly variable since 2013, with few LLINs manufactured since 2013 meeting WHO standards. This timeframe coincides with malaria resurgence in the country.	The abstract does not report actual WHO criteria for an acceptable net i.e. 80% bio efficacy OR 95% KD. It is unlikely that these results will be accepted by the community especially manufacturers. It is better to stick to the actual WHO thresholds throughout the text. While I appreciate the importance of the authors' point – using standard benchmarks are essential to conform with international standards.
Methods 125-126	WHO cone bioassays were conducted on the LLINs according to the standard protocol published by WHO [24], using either 50 or 25 (as recommended in [25]) fully pyrethroid susceptible Anopheles farauti mosquitoes either from a mosquito colony established at PNGIMR or collected locally as larvae and reared to adult stage. The experimental setup	More clarity is needed on the methods. For phase 3 (field studies) WHO advise 100 mosquitoes per net for unused nets and 80 for used nets.

	is shown in Figure 1.	4.7.3 Insecticidal activity The insecticidal activity (biological efficacy) of the nets should be determined in WHO cone tests and, when necessary, in tunnel tests (sections 2.2.1 and 2.2.2). Baseline bioassays should be conducted on five pieces of netting measuring 25 x 25 cm taken as shown in Figure 10, with 20 mosquitoes exposed on each piece of netting (total = 100 mosquitoes per net). In subsequent follow-ups, the piece removed from the bottom of the net (position 1) should be excluded, as it may be exposed to excessive abrasion if tucked under the bed (total = 80 mosquitoes per net). As in phase I studies, when knock-down is < 95% and mortality is < 80% on a given LN, it should be subjected to a tunnel test. For each net that fails to meet the criteria of the WHO cone test, the tunnel test should be conducted on the piece of netting that results in the mean mortality closest to that in the WHO cone assay. A sample table for presenting data on insecticidal activity is shown in Annex 13.
Methods 131		What is the evidence for susceptibility? Were the mosquitoes tested?
Methods 132		Were the mosquitoes sugar fed? What were the holding conditions for the 24hour observation period? Add these details.?
Methods 137-141	Of the LLINs tested, n=19 (including 1 positive and 1 negative control) were sent to Liverpool School of Tropical Medicine (LSTM) for confirmation of results through blinded repetition of the WHO cone bioassays with an identical Standard Operating Procedure and An. gambiae S3 standard pyrethroid susceptible laboratory strain.	Details are required. I am certain that the SOPs are not identical seeing as the site in PNG used ambient conditions and LSTM uses controlled conditions. I'm also interested to see how many mosquitoes were used per net sample.
Methods 142		The authors do not use means as WHO do for presenting the data. There is a justification for using the median. However, the authors then go on to use parametric statistics that assume a normal distribution. This is confusing, and possibly a Chi-squared test to assess the proportion of nets of each product passing the WHO bioefficacy criteria based on morality adjusted for control mortality and knock down, would be more appropriate.

Results 152-162		Strongly recommend the authors use conventional reporting first (proportion pass / fail on combined mortality / KD) and then go into the details of median mortality.
Results 184	Colony mosquitoes surviving the cone bioassays on these nets were able to blood feed in membrane feeds, oviposit and produce viable offspring.	An afterthought – no mention in the methods and no data to back up the statement. Include in detail or remove.
1 Results 94-196	The average 24h-mortality in the group of nets that was reported to have been in use between 1-3 years was 48% (95% CI 27%-69%), whereas average 24h mortality in the group of LLIN that was reported to have been in use for >3 years was 56% (95% CI 42%-71%).	Statistics section mentions that medians and IQR will be used.
Results 196-199	The proportion of LLINs with >80% 24h mortality in these two groups was 29% (4/14) and 38% (10/26), respectively. The proportion of LLINs resulting in 100% 24h mortality in the cone bioassays was 14% (2/14) and 19% (5/26) for these two groups, respectively.	Use conventional WHO criteria – pass fail on mortality OR KD60
Figure 2		Please use colour for figure 2C
Figure 3		Figure 3 better if combined pass / fail by year is presented. 100% mortality is not a WHO threshold

Discussion	Our present study suggests that it may be of benefit to recipient countries to implement this type of quality assessment to prevent distribution of LLINs with compromised bioefficacy and thus reduced utility for malaria control.	These results are extremely important. What do the authors recommend should be done? PNG has “quality control” in place. What more should be done?
------------	--	--

Minor

Line	Phrase	Comment
Throughout	Mins, hr	Ensure this is written in full on first use
29-30	In the present study, we observed a drastic reduction in bioefficacy of LLINs collected both from households as used nets and prior to use in original, unopened packaging.	In the present study, we observed a drastic reduction in bioefficacy of LLINs collected from households that had been used and nets that were unused in original, unopened packaging.
31-33	We hypothesise that decreased ability of LLINs to kill anopheline mosquitoes is a major contributor to the observed malaria resurgence in PNG and possibly in other parts of the world.	We evaluated the bioefficacy and chemical content to measure whether net quality had changed through time and may explain the observed malaria resurgence in PNG.
40-43		Add the actual number of nets that showed 100% mortality in addition to the percentage
64	However, malaria indicators have been on an upsurge in PNG in since 2015	However, malaria indicators have shown an upsurge in PNG in since 2015
65		State the current or last reported prevalence
108	found in Table 1 (large table at the end of the document).	found in Table 1.

123		Ref 25 is rather irrelevant as KD is an important feature of the mode of action of some nets.
152	WHO cone bioassays with new LLINs	WHO cone bioassays with unused LLINs
222-225	Our findings indicate that new LLINs distributed in PNG between 2013 and 2019 have not been exhibiting the required bioefficacy whereas LLINs distributed before 2013 performed significantly better [22].	Your data showed that they did not meet WHO requirements. Nets need to meet these requirements. Period.
264-252	It has been noted that the manufacturer made structural changes to the LLINs over the years. Bails of 100 LLINs weighing 50 kg in 2012 weighed as little as 43 kg in 2019, which translates to a weight reduction of 70 g per net. While these changes may be explained in terms of changing knitting weave, it calls into question if other changes have taken place and whether these LLINs have consequently been retested to confirm bioefficacy (information on average weight of LLIN bails vs year of manufacture can be found in Supporting Information Figure S2).	This is fascinating. Surely this means that these coated nets carry a little less insecticide. Would be great if this was discussed further.
257-259	A hypothesis arising from this is that while the LLINs may have the full concentration of insecticide the availability of insecticide may be restricted on the surface of the LLINs.	Why would this be the case for a coated net. I don't agree with the authors. It may be that too few samples are tested from throughout the entire consignment as there may be quality control issues with the nets leaving the manufacturers.

Table S1		I don't see what this adds to the paper

Comment 1: What do you mean by “multifactorial “?

‘The reasons for this resurgence are currently not well understood but are likely multifactorial.’

Comment 2: Please clarify: ‘WHO requirements for LLINs include that 80% of LLINs that have been in use for 3 years or less exhibit an 80% 24 h mortality...’

Comment 3: Materials and Methods section: For better presentation, suggest to have sub-heading for LLINs materials and details of each LLINs group should be placed under each respective sub-heading e.g. New LLINs (2018-2019): provide details...; LLINs (2007-2017): provide details; Used LLINs: provide details

Comment 4: Materials and Methods Section: Suggest to have details of three mosquito populations, where they were and etc

Comment 5: No information about the surface area (corner, middle or top) of the LLINs to be tested. Please specify if this has been done.

Comment 6: Which one is recommended by WHO(25 or 50)? Also how many replicates the authors have done and how many specimens were used per replicate

Comment 7: There are two sources of mosquito used in this study. Although both are susceptible to pyrethroids, there are considered different populations which may response to test compounds differently. What is the explanation in terms of geLLINic background?. Also which pyrethroids the authors referred to?

Comment 8: If the author knows the chemical name, please specify. From this point, discussion can be expanded.

Comment 9: The authors took small sample sites of LLINs between 2007 and 2012 (25) for this study (compared to those from 2013 to 2019 which included a large sample sites (167 LLINs). Please explain if the small sample sites provide sufficient information to make a conclusion.

Comment 10: Figure 2 caption: Explanation should be placed at the result section

Comment 11: No data “Colony mosquitoes surviving the cone bioassays on these LLINs were able to blood feed in membrane feeds, oviposit and produce viable offspring.”

Comment 12: Discussion: Authors need to address the mosquito behaviour in PNG with respect to vector control using LLINs. Any information on vector behaviour (e.g. endophilic, exophagic etc) has been reported in PNG. This is essential as LLINs

Point-by-point response

Decreased bioefficacy of long-lasting insecticidal nets and the resurgence of malaria in Papua New Guinea

We would like to thank the reviewers for the constructive and, overall, positive feedback.

Both reviewers agreed that the results we presented are very important. Point-by-point responses to each specific comment are provided below:

Reviewer 1

Comment 1: The malaria control community uses WHO guidance to test the quality of Insecticide treated nets. The authors need to revise their findings so that the results following WHO standard reporting are included i.e. the proportion of nets that pass or fail WHO bioefficacy criteria by year. This is important to allow comparison with other studies and to ensure that the reporting meets recognised standards.

Response 1: We have revised the presentation of the results to include the proportion of new LLIN that resulted in 80% 24h-mortality or 95% 60-min knockdown, which is the criterion outlined in the WHO guidelines [1]. This presentation is now part of Figures 2,3,4 and Table 1.

We would like to note that the WHO guidelines refer to phase 1,2 and 3 trials for testing candidate LLINs. Specifically, they are ‘[...] *intended to harmonize testing procedures in order to generate data for registration and labelling of [LLIN] products by national authorities and provide a framework for industry in developing novel [LLIN] products [...]*’ (page 1 in [1]). In the present study, we conducted a research involving an already certified product.

Comment 2: Authors should provide a table of results in addition to the multiple figures included in the manuscript. The table should include the number of nets tested by year, the number of mosquitoes tested % (95% CI) knock down % (95% CI) 24 hour mortality.

Response 2: We have included this table as requested. The new table and caption are shown below. I has been used to replace the previous Table 2.

Table 1: Summary of the number of LLINs and number of mosquitoes tested per year of manufacture. The table also shows the number of mosquitoes knocked down (KD) after 60 min and the number of mosquitoes dead after 24 h (dead), as well as the resulting % 60 min-knock-down (KD_{60min}) with 95% the exact confidence intervals of the proportions (95% CI) and % 24h-mortality (M_{24h}) with 95% CI.

Year	LLIN tested	mosquitoes tested	mosquitoes KD/dead	% KD_{60min} (95 % CI)	% M_{24h} (95 % CI)
2007	1	25	25/25	100 (86.28-100)	100 (86.28-100)
2008	3	75	69/74	92 (83.4-97.01)	98.67 (92.13-100)
2009	2	50	50/50	100 (92.89-100)	100 (92.89-100)
2010	7	175	173/173	98.86 (95.93-99.86)	98.86 (95.93-99.86)
2012	12	300	286/295	95.33 (92.29-97.43)	98.33 (96.15-99.46)
Sub-Total (pre-2013)	25	625	603/617	96.48 (94.72-97.78)	98.72 (97.49-99.45)
2013	20	500	243/241	48.6 (44.14-53.08)	48.2 (43.74-52.68)
2014	9	225	68/65	30.22 (24.3-36.68)	28.89 (23.06-35.29)
2015	24	600	339/369	56.5 (52.43-60.51)	61.5 (57.47-65.41)
2016	11	275	90/55	32.73 (27.21-38.62)	20 (15.44-25.22)
2017	54	1350	395/338	29.26 (26.84-31.77)	25.04 (22.75-27.44)
2018	27	662	396/385	59.82 (55.97-63.58)	58.16 (54.29-61.95)
2019	22	550	185/217	33.64 (29.69-37.76)	39.45 (35.35-43.68)
Sub-Total (2013-2019)	167	4162	1716/1670	41.23 (39.74-42.73)	40.12 (38.65-41.62)
Grand Total	192	4787	2319/2287	48.44 (47-49.9)	47.78 (46.4-49.2)

Comment 3: I am concerned that 2007 to 2010 have very small sample sizes. I would be happier if the nets from 2007 to 2012 were combined to one category for analysis and analysis was conducted on samples from before and after 2012.

Response 3: We have repeated the analysis as suggested by the reviewer, with test results separated into '2007-2012' and '2013-2019' groups. We present these results in the text as well as in the new table 1 (above), as well as Panel B of the revised Figure 3.

Comment 4: I do however like very much the data presentation that the authors have used, especially in figure 2 and the additional analysis should be included just to ensure that conventional reporting is also included for completeness and improved acceptability to all members of the malaria community, in particular manufacturers who should not be left with any grounds to contest the results.

Response 4: In response to this comment we have modified Figure 3 to show the proportions of LLINs per year with $\geq 95\%$ knock down or $\geq 80\%$ mortality (Panel A). The figure also has a panel B showing the results combined for '2007-2012' and '2013-2019' LLINs as requested in Comment 3. The new figure and figure caption are shown below. We have left the data reporting 100% mortality and 100% knockdown mentioned in the text.

Revised Figure 3: Proportion of unused LLIN exhibiting >95% 60min-knockdown (KD_{60min}) and >80% 24h-mortality (M_{24h}). Panel A: Proportions of LLINs are shown by year of manufacture. **Panel B:** Proportions of LLINs are shown for groups with years of manufacture between 2007-2012 (pre-2013) and from 2013 to 2019, as well as overall (black circle). The table on top of the panels indicates the numbers of LLINs tested in each year or group. The error bars are exact 95% confidence intervals of proportions. The p-value between the pre-2013 and 2013-2019 groups is the result of a chi squared test comparing the proportions in these two groups.

Comment 5: The authors need to spend some time editing the methods section as it is not possible to follow exactly what was done as written. I have provided more detailed suggestions in the tables. Once this is done it may be possible to evaluate whether the correct WHO procedures were performed. If they were not followed then the results must be justified to ensure that sufficient replication per net was used.

Response 5: We have added as much detail as possible to the methods section. The changes are highlighted with track changes in the manuscript. This section now reads:

WHO cone bioassays

WHO cone bioassays were conducted on the LLINs according to WHO guidelines [1], using 25 (as recommended in [2]) fully pyrethroid susceptible Anopheles farauti mosquitoes.

The experimental setup of the WHO cone bioassays is shown in Figure 1.

Mosquitoes were 3-5 days old when used in the cone bioassays. Tests were conducted in ambient tropical environment (Madang, PNG, latitude 5° south) and temperature and humidity requirements were met in all assays included in the study. The number of mosquitoes per cone was n=5 and exposure time to the LLINs was 3 minutes (min).

All cone bioassays included positive and negative controls. We used LLINs manufactured in 2012 and with a known 100% 24h mortality as positive controls and pieces of untreated netting as negative controls. Results were excluded if 24h mortality in the negative control exceeded 10%. Tests results were adjusted using 'Abbott's formula' when negative control 24h mortality was > 0% and <10%.

After exposure to the LLINs, mosquitoes were gently transferred from the cones to cardboard holding cups screened with untreated netting and provided access to 10% sugar solution via a soaked piece of cotton wool placed on top of the netting. After 60 min, the number of mosquitoes knocked down in the holding cups was enumerated and after 24h, the number of dead mosquitoes in the holding cups was enumerated.

The total number of mosquitoes exposed to each LLIN section was n=25 in 5 cones and one LLIN side panel section (30 x 30 cm) was tested per LLIN as shown in Figure 1, placing the cones onto different parts of the section. This approach is different to WHO guidelines for large scale (phase 3) LLIN candidate product evaluation where n=5 sections from n=30 unused LLINs are tested with n=20 mosquitoes each (i.e., n=100 mosquitoes per LLIN). The reason this different approach was selected is that we aimed to test a much larger number of

LLIN samples from an already WHO certified product to span as many years and batches as possible.

Of the LLINs tested, n=19 (including 1 positive and 1 negative control) were sent to Liverpool School of Tropical Medicine (LSTM) for confirmation of results through blinded repetition of the cone bioassays with a very similar Standard Operating Procedure based on WHO guidelines and the An. gambiae G3 standard pyrethroid susceptible laboratory strain. The LSTM laboratory used n=50 mosquitoes to conduct the repetitions (i.e., 10 cones with 5 mosquitoes each).

In addition to the modifications to the manuscript text above, we would like to point out that the main aim of this study was not to conduct a phase 3 LLIN trial as required for WHO-certification of a new product. Instead, the aim was to conduct a research study on LLIN bioefficacy in PNG to understand if malaria resurgence there could be, in part, a result of a change in LLIN bioefficacy. As such it is reasonable that some of the selected methods are different.

In a phase 3 trial, the WHO guidelines state that a total of n=30 LLINs are to be tested at baseline (prior to use), and for each LLIN, 5 cuttings are exposed to n=20 mosquitoes (a total of 100 mosquitoes per LLIN). Overall this results in the exposure of n=3000 mosquitoes to LLIN material at baseline. In our study we tested n=192 LLINs i.e., a much higher number than what would be required in a phase 3 trial at baseline. These LLINs had n= 78 different batch numbers spanning 12 years. We tested one cutting per LLIN, which was consistently from one of the side panels, with n=25 mosquitoes. Overall, using our approach resulted in the exposure of a total of 4787 mosquitoes to the unused LLINs. As such our sample size is much larger for both, the number of LLINs tested and also the overall number of mosquitoes exposed.

Our results are overwhelmingly statistically significant and the effect size is large. For the n=25 LLINs from the pre-2013 group, 625 mosquitoes were exposed of which 603 (96%) were knocked down after 60 min and 617 (98%) were dead after 24 h. In contrast, for LLINs in the 2013 to 2019 group, a total of n=4162 mosquitoes were exposed to n= 167 LLINs and 1716 (41%) were knocked down after 60 min and 1670 (40%) were dead after 24h. As such, the observed effect size is large.

Consequently, when applying a chi-squared test to these data, we obtain χ^2 of 664.14 ($p < 10^{-5}$) for 60 min-knockdown and 747.75 ($p < 10^{-5}$) for 24 h-mortality, respectively, which is overwhelmingly statistically significant. Alternatively, when using univariate analysis of variance on the 192 LLIN grouped into pre-2013 and 2013-2019 categories, the estimated statistical power to detect the observed effect size is 100% for both, KD and mortality.

However, we have also conducted more experiments to address this comment, n=1 LLIN from each of the years 2007, 2008, 2009, 2010 and 2012 were tested again, this time using n=5 cuttings (four side panels, different heights, as well as the roof panels) with n=20 mosquitoes. In all these assays (n=25), for which the results are shown below, mosquito knockdown and mortality were 100%. Also, we have consistently used LLINs from 2012 as positive controls (as all of them kill 100% of mosquitoes). As such a large number of 2012 LLIN cuttings was tested in the form of positive controls.

Table 2: Additional results from 2007-2012 LLINs. BS-bottom side, MS-middle side and TS-top side. The numbers 1-5 indicate the positions from which netting pieces were cut, i.e; according to the WHO Guidelines for laboratory and field-testing of LLINs.

Brand	Net ID	Panel	Year of Man.	KD (%)	Mortality (%)
PermaNet 2.0	EHP 046 BS#1	bottom side	2007	100	100
PermaNet 2.0	EHP 046 MS#2	middle side	2007	100	100
PermaNet 2.0	EHP 046 Roof#3	roof	2007	100	100
PermaNet 2.0	EHP 046 MS#4	middle side	2007	100	100
PermaNet 2.0	EHP 046 TS#5	top side	2007	100	100

PermaNet 2.0	CHI 039 BS#1	bottom side	2008	100	100
PermaNet 2.0	CHI 039 MS#2	middle side	2008	100	100
PermaNet 2.0	CHI 039 Roof#3	roof	2008	100	100
PermaNet 2.0	CHI 039 MS#4	middle side	2008	100	100
PermaNet 2.0	CHI 039 TS#5	top side	2008	100	100
				100	
PermaNet 2.0	KAV 003 BS#1	bottom side	2009	100	100
PermaNet 2.0	KAV 003 MS#2	middle side	2009	100	100
PermaNet 2.0	KAV 003 Roof#3	roof	2009	100	100
PermaNet 2.0	KAV 003 MS#4	middle side	2009	100	100
PermaNet 2.0	KAV 003 TS#5	top side	2009	100	100
				100	
PermaNet 2.0	EHP 029 BS#1	bottom side	2010	100	100
PermaNet 2.0	EHP 029 MS#2	middle side	2010	100	100
PermaNet 2.0	EHP 029 Roof#3	roof	2010	100	100
PermaNet 2.0	EHP 029 MS#4	middle side	2010	100	100
PermaNet 2.0	EHP 029 TS#5	top side	2010	100	100
				100	
PermaNet 2.0	EHP 031 BS#1	bottom side	2012	100	100
PermaNet 2.0	EHP 031 MS#2	middle side	2012	100	100
PermaNet 2.0	EHP 031 Roof#3	roof	2012	100	100
PermaNet 2.0	EHP 031 MS#4	middle side	2012	100	100
PermaNet 2.0	EHP 031 TS#5	top side	2012	100	100

Based on the magnitude of the observed difference, the overwhelming statistical significance and the large number of LLIN tested and the large number of mosquitoes exposed, and the confirmatory data from the additional experiments, we trust that our results are conclusive even though we did not routinely test multiple cuttings per individual LLIN. We have added some of this rationale to the discussion section of the manuscript.

Comment 6: The study has a number of limitations such as the small sample size and the unknown storage conditions

Response 6: We acknowledge that n=25 LLINs prior to 2013 is a relatively small number – as highlighted by the reviewer, it is quite difficult to obtain unused nets in original packaging from 8-10 years ago. We have added a paragraph in the discussion section to mention these limitations. The added text is:

'Although the number of LLIN tested for the 2007-2012 period (n=25) is relatively small due to the difficulty of obtaining unused LLINs from such a long time ago, the difference between the 2007-2012 and the 2013-2019 groups was large and overwhelmingly statistically significant (chi squared equal to 75.4; p-value <10⁻⁵).'

However, we do not consider the overall sample size of 192 unused LLINs as small. In comparison, in a complete large scale phase 3 trial n=230 LLINs would be tested for bioefficacy [1].

We would also like to point out that one of the most important results of our study is not the difference between 2007-2012 LLIN and 2013-2019 LLIN groups, but the observation that most unused LLIN tested in this study did not kill 100% (or 80%) of susceptible Anopheles mosquitoes or achieved 100% (or 95%) knock down. As such, even if the n=25 sample size of 2007-2012 LLINs is considered small and the overwhelming statistical evidence summarized in the previous paragraph is put aside, we have still shown on a large number of unused LLINs (n=192) that most of these LLINs do not kill mosquitoes as expected.

In the following, we respond to each comment in the additional ‘response’ column in the table below.

Major Comments

Line	Phrase	Comment	Response
Intro 46-48	Bioefficacy of LLINs in PNG appears to have been highly variable since 2013, with few LLINs manufactured since 2013 meeting WHO standards. This timeframe coincides with malaria resurgence in the country.	The abstract does not report actual WHO criteria for an acceptable LLIN i.e. 80% bio efficacy OR 95% KD. It is unlikely that these results will be accepted by the community especially manufacturers. It is better to stick to the actual WHO thresholds throughout the text. While I appreciate the importance of the authors’ point – using standard benchmarks are essential to conform with international standards.	We have revised the entire abstract to adhere to the 150 word limit. We have included the analysis as requested by the reviewer and are now reporting the proportion of LLINs that resulted in $\geq 80\%$ mortality or $\geq 95\%$ knockdown. We have updated all tables and figures, as well as the text in order to reflect these changes..
Methods 125-126	WHO cone bioassays were conducted on the LLINs according to the standard protocol published by WHO [24], using either 50 or 25 (as recommended in [25]) fully pyrethroid susceptible Anopheles farauti mosquitoes either from a mosquito colony established at PNGIMR or collected locally as larvae and reared to adult stage. The experimental setup is shown in Figure 1.	More clarity is needed on the methods. For phase 3 (field studies) WHO advise 100 mosquitoes per LLIN for unused LLINs and 80 for used LLINs.	We apologise for the lack of clarity and have revised the methods section to include more detail on the methods used in this study. We believe that there is now sufficient detail to enable an exact repetition of the experiments. After reviewing the data, we realised that only preliminary experiments on some of the used nets, as well as the confirmatory experiments at LSTM were conducted with 50 mosquitoes. All other results included in this study were obtained using n=25 mosquitoes and 1 cutting per LLIN resulting in exposure of almost 4782 mosquitoes to test unused LLINs and 1626 mosquitoes to used LLIN. We acknowledge that for the results presented on used LLINs, this is a limitation, as the lower

			part of the LLINs can be subject to higher rate of wear and tear. We have now included a statement with respect to this in the discussion section. For unused LLIN we see no reason to assume a different insecticide content in the different side panels or the roof and even if this was the case, we are confident that this would have no effect on the main conclusions of our study. It has been shown that using 25 mosquitoes per test is sufficient to obtain accurate results for 24h-mortality and very reasonable estimates for 60 min-knock down as compared to n=50 mosquitoes. This was not a phase 3 study but a research study that aimed at including as many LLINs as possible. As such, it included a much higher number of LLIN then a phase 3 study at baseline (192 LLINs in our study vs 30 LLINs in a phase 3 study at baseline), as well as a higher number of exposed mosquitoes.
Methods 131		What is the evidence for susceptibility? Were the mosquitoes tested?	We regularly test colony mosquitoes and wild mosquitoes from the collection area using WHO tube assays. This included tests before, during and after the present study. We consistently observe 100% mortality and knock-down with colony and wild-type mosquitoes against all pyrethroids tested and this includes deltamethrin. We have recently obtained results using CDC bottle-assays for

			deltamethrin for both types of mosquitoes used in this study as well that support these results. We have added a statement regarding this to the methods section. There is also a publication from 2018 with wild mosquito testing results from our work in PNG showing no evidence of pyrethroid resistance in wild populations across the country. As of today, we have not found any sign of pyrethroid resistance in PNG.[3]
Methods 132		Were the mosquitoes sugar fed? What were the holding conditions for the 24hour observation period? Add these details.?	Yes- the mosquitoes were provided with 10% sugar solution in the form of soaked cotton balls placed on the top of the netting on the holding cups. We have added this detail in the methods section.
Methods 137-141	Of the LLINs tested, n=19 (including 1 positive and 1 negative control) were sent to Liverpool School of Tropical Medicine (LSTM) for confirmation of results through blinded repetition of the WHO cone bioassays with an identical Standard Operating Procedure and An. gambiae S3 standard pyrethroid susceptible laboratory strain.	Details are required. I am certain that the SOPs are not identical seeing as the site in PNG used ambient conditions and LSTM uses controlled conditions. I'm also interested to see how many mosquitoes were used per LLIN sample.	We have added more details on the experimental setup used at LSTM. The reviewer is correct, the word 'identical' was not well-chosen. We have reworded this sentence stating that the methods were 'very similar' and provide the details of the differences.

Methods 142		The authors do not use means as WHO do for presenting the data. There is a justification for using the median. However, the authors then go on to use parametric statistics that assume a normal distribution. This is confusing, and possibly a Chi-squared test to assess the proportion of LLINs of each product passing the WHO bioefficacy criteria based on mortality adjusted for control mortality and knock down, would be more appropriate.	We have rectified this and have taken the reviewers suggestion regarding the chi squared test on board. Since these are not normally distributed data, it should really be medians that are presented and we have ensured that this is done throughout the text. The data are adjusted to negative control mortality and knock-down using Abbott's formula and we have added this statement in the methods section.
Results 152-162		Strongly recommend the authors use conventional reporting first (proportion pass / fail on combined mortality / KD) and then go into the details of median mortality.	We have changed the presentation of the results to adhere to this suggestion. We replaced Figure 3 with a new Figure showing these data.
Results 94-196	The average 24hmortality in the group of LLINs that was reported to have been in use between 1-3 years was 48% (95% CI 27%- 69%), whereas average 24h mortality in the group of LLIN that was reported to have been in use for >3 years was 56% (95% CI 42%- 71%).	Statistics section mentions that medians and IQR will be used.	We have corrected this. Since these are not normally distributed data, it should really be medians that are presented and we have ensured that this is done throughout the text
Results 196-199	The proportion of LLINs with >80% 24h mortality in these two groups was 29% (4/14) and 38% (10/26),	Use conventional WHO criteria – pass fail on mortality OR KD60	We have changed the presentation of the results to adhere to this advice

	respectively. The proportion of LLINs resulting in 100% 24h mortality in the cone bioassays was 14% (2/14) and 19% (5/26) for these two groups, respectively.		
Figure 2		Please use colour for figure 2C	We have provided an updated version of the Figure using different colours and shapes to separate the data by year of manufacture (2007-2012: green; 2013-2019: orange). We applied this colour coding throughout the manuscript.
Figure 3		Figure 3 better if combined pass / fail by year is presented. 100% mortality is not a WHO threshold	We have developed a new Figure 3 in which the data are presented as suggested by the reviewer
Discussion	Our present study suggests that it may be of benefit to recipient countries to implement this type of quality assessment to prevent distribution of LLINs with compromised bioefficacy and thus reduced utility for malaria control.	These results are extremely important. What do the authors recommend should be done? PNG has “quality control” in place. What more should be done?	PNG has relied on pre-delivery inspections which look at insecticide content and, more recently, wash ability. The LLINs tested in this study passed these inspections. However, it would appear that these inspections give little information about bioefficacy. As such, bioefficacy testing should be included in the pre-delivery inspections. Ideally, recipient countries should be enabled to carry out these inspections themselves. Alternatively, global capacity to carry out bioefficacy tests should be reviewed and there should be a number of reference laboratories that countries can turn to in case faulty nets are suspected. In PNG, this study was only possible due to the good collaboration between PNGIMR and the national malaria control program.

			We have expanded the discussion surrounding this.
--	--	--	---

Minor Comments

Line	Phrase	Comment	Response
Throughout	Mins, hr	Ensure this is written in full on first use	We have reviewed the text for consistency
29-30	In the present study, we observed a drastic reduction in bioefficacy of LLINs collected both from households as used LLINs and prior to use in original, unopened packaging.	In the present study, we observed a drastic reduction in bioefficacy of LLINs collected from households that had been used and LLINs that were unused in original, unopened packaging.	We accepted this suggested edit.
31-33	We hypothesise that decreased ability of LLINs to kill anopheline mosquitoes is a major contributor to the observed malaria resurgence in PNG and possibly in other parts of the world.	We evaluated the bioefficacy and chemical content to measure whether LLIN quality had changed through time and may explain the observed malaria resurgence in PNG.	We have not measured chemical content in our study. We have access to the quality assessments from Crown Agents and TUV SUD and they suggest that the chemical content was within the specified range
40-43		Add the actual number of LLINs that showed 100% mortality in addition to the percentage	We have included the number of LLINs in the figure.
64	However, malaria indicators have been on an upsurge in PNG in since 2015	However, malaria indicators have shown an upsurge in PNG in since 2015	We accepted this suggested edit.
65		State the current or last reported prevalence	We have included the latest estimate in the manuscript

108	found in Table 1 (large table at the end of the document).	found in Table 1.	We accepted this suggested edit.
123		Ref 25 is rather irrelevant as KD is an important feature of the mode of action of some LLINs.	In reference 25 it was shown that using 25 mosquitoes per LLIN results in robust 24h mortality estimates. As such it is an important reference to cite as we have used 25 mosquitoes per net. In the same paper, it is shown that the combined KD/mortality estimate using 25 mosquitoes has a sensitivity of 87.6% and 90.6% and a specificity between 89.9% and 96.3% when compared to the method outlined in the WHO guidelines.
152	WHO cone bioassays with new LLINs	WHO cone bioassays with unused LLINs	We accepted this suggested edit.
222-225	Our findings indicate that new LLINs distributed in PNG between 2013 and 2019 have not been exhibiting the required bioefficacy whereas LLINs distributed before 2013 performed significantly better [22].	Your data showed that they did not meet WHO requirements. LLINs need to meet these requirements. Period.	We agree with this statement.
264-252	It has been noted that the manufacturer made structural changes to the LLINs over the years. Bails of 100 LLINs weighing 50 kg in 2012 weighed as little as 43 kg in 2019, which translates to a weight	This is fascinating. Surely this means that these coated LLINs carry a little less insecticide. Would be great if this was discussed further.	The pre-distribution inspections conducted by independent agencies (Crown Agents/TUV SUD) indicate that the insecticide content in these LLINs is similar over the years. However, it is true that potentially there may be a proportional reduction in insecticide per sqm simply because there is less material. Average LLIN weight decreased by around

	reduction of 70 g per LLIN. While these changes may be explained in terms of changing knitting weave, it calls into question if other changes have taken place and whether these LLINs have consequently been retested to confirm bioefficacy (information on average weight of LLIN bails vs year of manufacture can be found in Supporting Information Figure S2).		15%. We consider it unlikely that a proportional decrease in insecticide of 15% across the surface area would result in the dramatic effect that have observed.
257-259	A hypothesis arising from this is that while the LLINs may have the full concentration of insecticide the availability of insecticide may be restricted on the surface of the LLINs.	Why would this be the case for a coated LLIN. I don't agree with the authors. It may be that too few samples are tested from throughout the entire consignment as there may be quality control issues with the LLINs leaving the manufacturers.	While we in principle agree with this statement, since we do not know the exact manufacturing process, we cannot exclude this possibility. We do not know the influence of the binding agent fusing the insecticide to the net surface or whether there is an additional layer of material applied after coating with the insecticide. The chemical analyses by independent agencies (Crown Agents and TUV SUD) suggest that the chemical content is within the specified range.
Table S1		I don't see what this adds to the paper	We agree that this table is not immediately relevant to the conclusions in this paper. We have deleted Table S1

Reviewer 2

Comment 1: What do you mean by “multifactorial “?

Response 1: The reviewer refers to this statement on page ... in the introduction section:

‘The reasons for this resurgence are currently not well understood but are likely multifactorial.’

By ‘multifactorial’ we mean that the reasons for the resurgence of malaria in PNG could be due to multiple reasons and is most likely a result of multiple factors acting together. These factors are described in the text immediately following this statement. We outline and discusses several potential reasons for the observed malaria resurgence in PNG: 1) drug stock outs; 2) vector behaviours; 3) insecticide resistance.

In response to the reviewer comment we changed the wording to: *‘The cause of this resurgence is currently not well understood but is likely due to multiple factors acting together.’*

Comment 2: Reviewer 2 asks for clarification of the following statement: ‘WHO requirements for LLINs include that 80% of LLINs that have been in use for 3 years or less exhibit an 80% 24 h mortality...’

Response 2: This is part of the background section, where we refer to the WHO guidelines [1] pertaining to the testing of candidate LLIN products for certification: ‘A candidate LLIN is considered to meet the criteria for efficacy for testing in phase III studies if, after 3 years, at least 80% of sampled LLINs are effective in WHO cone tests ($\geq 95\%$ knockdown or $\geq 80\%$ mortality)...’

We have revised the statement as follows: *‘According to WHO guidelines, LLINs that have been in use for 3 years or less should exhibit 80% 24 h mortality or 95% 60 min-knock down in standardised WHO cone bioassays’*

Comment 3: Materials and Methods section: For better presentation, suggest to have sub-heading for LLINs materials and details of each LLINs group should be placed under each respective sub-heading e.g. New LLINs (2018-2019): provide details...; LLINs (2007-2017): provide details; Used LLINs: provide details

Response 3: We have revised the section according to the reviewer suggestion. I now reads:

‘Sources of LLINs tested in the present study were as follows

Unused LLINs manufactured in 2018 and 2019 (n=): These LLINs were provided by RAM PNG from consignments dedicated to different PNG provinces

Unused LLINs manufactured in 2007 to 2017 (n=25): These LLINs were obtained from villages or provincial health authorities in various PNG provinces. All LLINs were still in original and unopened packaging,

Overall, n=192 unused LLINs with 78 different batch numbers were tested. These had been distributed to or were intended for distribution in 15 PNG Provinces, namely Central, Chimbu, East New Britain, East Sepik, Eastern Highlands, Gulf, Hela, Manus, Morobe, New Ireland, Oro, Southern Highlands, Western, Western Highlands and West New Britain.

Details on the LLINs tested can be found in Table 1.

Used LLINs (n=40): These LLINs were collected in communities in Madang Province and Central Province in 2018 and 2019, and owners were asked to indicate how long they had been using these LLINs. These LLINs showed signs of wear and were not in original packaging.’

Comment 4: Materials and Methods Section: Suggest to have details of three mosquito populations, where they were and etc

Response 4: We have used colony *An. farauti* mosquitoes and wild-type *An. farauti* mosquitoes in the present study (2 populations). We have revised the section according to the reviewer suggestion, providing more details on these mosquito populations. Specifically we have added:

'Specifically, the laboratory colony used in the present study was an An. farauti strain originally from Rabaul, East New Britain province, PNG. This colony has existed for many decades and was back-crossed with local An. farauti mosquitoes from Madang province in PNG several times, the last time in 2012. Regular WHO-tube testing on the colony mosquitoes before, throughout and after completion of this study confirmed susceptibility to deltamethrin, lambda-cyhalothrin, DDT, bendiocarb and malathion insecticides. Local wild-caught Anopheles farauti mosquitoes used in this study originated from larval collections along the North Coast of Madang Province, PNG. Extensive insecticide resistance monitoring using WHO tube assays was conducted during the same time and with larvae collected from the same habitats confirming full susceptibility of these wild-caught mosquitoes to deltamethrin, lambda-cyhalothrin, DDT, Bendiocarb and Malathion insecticides. Overall, there is no indication of pyrethroid resistance in Anopheline populations from anywhere in PNG.' [3]

Comment 5: No information about the surface area (corner, middle or top) of the LLINs to be tested. Please specify if this has been done.

Response 5: We have revised the methods section providing this information. Specifically, we have added the following text:

‘The total number of mosquitoes exposed to each LLIN section was n=25 and one LLIN side panel section (30 x 30 cm) was tested per LLIN as shown in Figure 1. This approach is different to WHO guidelines for large scale (phase 3) LLIN product evaluation where n=5 sections from n=30 unused LLINs are tested with n=20 mosquitoes each (i.e., n=100 mosquitos per LLIN). The reason we chose this different approach is that we aimed to test a much larger number of LLIN samples from an already WHO certified product. In this study we tested n=192 unused LLIN in contrast to the required n=30 unused LLIN in a phase 3 trial.’

Comment 6: Which one is recommended by WHO(25 or 50)? Also how many replicates the authors have done and how many specimens were used per replicate

Response 6: We have added these details to the methods section. Specifically, we have added the following text:

‘The total number of mosquitoes exposed to each LLIN section was n=25 in 5 cones (5 mosquitoes per cone) and one LLIN side panel section (30 x 30 cm) was tested per LLIN as shown in Figure 1, placing the cones onto different parts of the section. This approach is different to WHO guidelines for large scale (phase 3) LLIN product evaluation where n=5 sections from n=30 unused LLINs are tested with n=20 mosquitoes each (i.e., n=100 mosquitos per LLIN). The reason we chose this different approach is that we aimed to test a much larger number of LLIN samples from an already WHO certified product to span as many years and batches as possible. In this study we tested n=192 unused LLIN in contrast to the required n=30 unused LLIN in a phase 3 trial.’

Comment 7: There are two sources of mosquito used in this study. Although both are susceptible to pyrethroids, there are considered different populations which may response to test compounds differently. What is the explanation in terms of geLLINic background?. Also which pyrethroids the authors referred to?

Response 7: Yes, both susceptible colony and susceptible wild-type mosquitoes were used in this study. We have very solid, regular tube-assay testing data for both populations confirming their susceptibility to deltamethrin, lambda-cyhalothrin, DDT, bendiocarb and malathion. The tube-assay results from these two populations are always 100% mortality and 100% knockdown for deltamethrin.[3] The colony has been back-crossed with local *Anopheles farauti* mosquitoes twice, the last time in 2012. Please see our response to comment 4 for the details added to the methods section describing these mosquito populations.

Comment 8: If the author knows the chemical name, please specify. From this point, discussion can be expanded.

Response 8: We changed the insecticide specification from 'pyrethroid' to 'deltamethrin' where we considered this appropriate.

Comment 9: The authors took small sample sites of LLINs between 2007 and 2012 (25) for this study (compared to those from 2013 to 2019 which included a large sample sites (167 LLINs). Please explain if the small sample sites provide sufficient information to make a conclusion.

Response 9: As pointed out by Reviewer 1 it is quite difficult to locate new/unused LLINs from such a long time ago. In the present study, they were located in villages where people had kept them within the original unopened packaging for (in case of 2007 and 2008) more than 10 years. We managed to locate 25 such nets. However, we are confident that n=25 LLINs is a sufficient number as the effect size of the difference that we are observing is large. As per our response to a similar comment from Reviewer 1, from a statistical perspective, our results are very solid. For the n=25 LLINs from the pre-2013 group, 625 mosquitoes were exposed of which 603 (96%) were knocked down after 60 min and 617 (98%) were dead after 24 h. In contrast, for LLINs in the 2013 to 2019 group, a total of n=4162 mosquitoes were exposed to n= 167 LLINs and 1716 (41%) were knocked down after 60 min and 1670 (40%) were dead after 24h. As such, the observed effect size is large.

Consequently, when applying a chi-squared test to these data, we obtain χ^2 of 664.14 ($p < 10^{-5}$) for 60min-knockdown and 747.75 ($p < 10^{-5}$) for 24h-mortality, respectively, which is extremely statistically significant. Alternatively, when using univariate analysis of variance on the 192 LLIN grouped into pre-2013 and 2013-2019 categories, the estimated observed statistical power to detect the observed effect size is 100% for both, KD and mortality.

We have addressed this further in the discussion section by adding the following text:

‘Although the number of LLIN tested for the 2007-2012 period was relatively small due to the difficulty of obtaining unused LLINs from such a long time ago, the difference between the 2007-2012 and the 2013-2019 groups was large and overwhelmingly statistically significant (chi squared equal to 75.4; p-value <10-5).’

Comment10 : Figure 2 caption: Explanation should be placed at the result section

Response 10: It is our understanding that a figure caption should allow the reader to understand the figure content without referring back to the text if possible. As such, the details we are providing in the captions are needed. We will seek editorial advice on this and will modify the captions placing the details into the text if requested to do so by the editor.

Comment 11: The reviewer noted that we present no data associated with the statement: Colony mosquitoes surviving the cone bioassays on these LLINs were able to blood feed in membrane feeds, oviposit and produce viable offspring.

Response 11: We agree that we do not have enough systematic data to support this statement and have deleted it.

Comment 12: Discussion: Authors need to address the mosquito behaviour in PNG with respect to vector control using LLINs. Any information on vector behaviour (e.g.endophilic, exophagic etc) has been reported in PNG. This is essential as LLINs

Response 12: This comment of the reviewer appears to be cut short. We are describing PNG Anopheline vector behaviour on page 3 (introduction section). We have expanded this description. Specifically we added the text:

‘Vector behavioural patterns may also determine LLIN impact. In general, PNG malaria vectors, including the main coastal vector Anopheles farauti tend to be exophagic and, with few exceptions, exhibit opportunistic host preferences [4, 5]. These behavioural patterns are believed to be detrimental to LLIN impact as most human-vector contact occurs outside the house and vectors can easily seek alternative hosts. In addition, studies have also suggested behavioural adaptation of local malaria vectors to LLINs, such that biting now occurs earlier in the evening [6, 7]’

References

1. WHO: **Guidelines for laboratory and field testing of long-lasting insecticidal nets.** (WHO ed. Geneva, Switzerland: WHO Document Production Services; 2013.
2. Boyer S, Pothin E, Randriamaherijaona S, Rogier C, Kesteman T: **Testing bio-efficacy of insecticide-treated nets with fewer mosquitoes for enhanced malaria control.** *Sci Rep* 2018, **8**:16769.
3. Koimbu G, Czeher C, Katusele M, Sakur M, Kilepak L, Tandrapah A, Hetzel MW, Pulford J, Robinson L, Karl S: **Status of Insecticide Resistance in Papua New Guinea: An Update from Nation-Wide Monitoring of Anopheles Mosquitoes.** *Am J Trop Med Hyg* 2018, **98**:162-165.
4. Keven JB, Katusele M, Vinit R, Koimbu G, Vincent N, Thomsen EK, Karl S, Reimer LJ, Walker ED: **Species abundance, composition, and nocturnal activity of female Anopheles (Diptera: Culicidae) in malaria-endemic villages of Papua New Guinea: assessment with barrier screen sampling.** *Malar J* 2019, **18**:96.
5. Keven JB, Reimer L, Katusele M, Koimbu G, Vinit R, Vincent N, Thomsen E, Foran DR, Zimmerman PA, Walker ED: **Plasticity of host selection by malaria vectors of Papua New Guinea.** *Parasit Vectors* 2017, **10**:95.
6. Reimer LJ, Thomsen EK, Koimbu G, Keven JB, Mueller I, Siba PM, Kazura JW, Hetzel MW, Zimmerman PA: **Malaria transmission dynamics surrounding the first nationwide long-lasting insecticidal net distribution in Papua New Guinea.** *Malar J* 2016, **15**:25.
7. Thomsen EK, Koimbu G, Pulford J, Jamea-Maiasa S, Ura Y, Keven JB, Siba PM, Mueller I, Hetzel MW, Reimer LJ: **Mosquito behaviour change after distribution of bednets results in decreased protection against malaria exposure.** *J Infect Dis* 2016.

REVIEWERS' COMMENTS:

Reviewer #1 (Remarks to the Author):

The resubmitted manuscript is excellent. It reads well, the figures are great and the authors answered all of my comments thoroughly. The grouping of the data pre and post 2012 is extremely helpful making the result very clear and the conventional reporting allows the community to see that the majority of nets fail international standards after 2012, and action to understand the underlying causes of this observation is required.

I recommend publication with the correction of a few minor items.

1. One small error on my part - the more appropriate guidance for this kind of study is WHO (2011). Guidelines for monitoring the durability of long-lasting insecticidal mosquito nets under operational conditions. Geneva, World Health Organization

This recommends 40 mosquitoes per net.

Therefore, I take your point in comment 1 (this is not a new registration) and refer you to the guidance on monitoring, which is essentially what you have done. I would use this reference instead of ref 26 as it is the correct one, and I accept your justification for selection of fewer mosquitoes based on Boyer 2018. My apology for that oversight. I suggest you update the methods and discussion to reflect the use of operational monitoring guidance. However, changing the reference does not affect the suggestions on data reporting and I am glad that you have used conventional benchmarks for comparability with other studies produced by the research community.

2. The reference section has some errors and needs fixing.

3. Abstract

We hypothesise that decreased bioefficacy of LLINs is a major contributor to the malaria resurgence in PNG.

I would use a slightly stronger statement "These results suggest that decreased bioefficacy of LLINs is contributing to the malaria resurgence in PNG and increased scrutiny of LLIN quality is warranted."

4. Discussion

This paper appears to have uncovered a potentially big problem in net quality since so many batches failed over such a large time frame. I would like to see the authors make a stronger recommendation in the discussion and call for quality assurance through operational monitoring of nets in all endemic countries to ensure that this is not a global problem.

Finally - well done - this is excellent work of high importance that should be published immediately.

Point-by-Point Response to Reviewer Comments

Decreased bioefficacy of long-lasting insecticidal nets and the resurgence of malaria in Papua New Guinea

We would like to thank the reviewer once again for the supportive comments and overall positive feedback. Below, we address each specific comment raised by the reviewer.

General Comment

The resubmitted manuscript is excellent. It reads well, the figures are great and the authors answered all of my comments thoroughly. The grouping of the data pre and post 2012 is extremely helpful making the result very clear and the conventional reporting allows the community to see that the majority of nets fail international standards after 2012, and action to understand the underlying causes of this observation is required. I recommend publication with the correction of a few minor items.

Specific Comments:

Comment 1: One small error on my part - the more appropriate guidance for this kind of study is WHO (2011). Guidelines for monitoring the durability of long-lasting insecticidal mosquito nets under operational conditions. Geneva, World Health Organization

This recommends 40 mosquitoes per net. Therefore, I take your point in comment 1 (this is not a new registration) and refer you to the guidance on monitoring, which is essentially, what you have done. I would use this reference instead of ref 26 as it is the correct one, and I accept your justification for selection of fewer mosquitoes based on Boyer 2018. My apology for that oversight. I suggest you update the methods and discussion to reflect the use of operational monitoring guidance. However, changing the reference does not affect the suggestions on data reporting and I am glad that you have used conventional benchmarks for comparability with other studies produced by the research community.

Response 1: Thank you for this comment. We have included the aforementioned reference 'WHO (2011). Guidelines for monitoring the durability of long-lasting insecticidal mosquito nets under operational conditions. Geneva, World Health Organization' in Methods and Discussion sections.

Specifically, we have added the text:

Discussion: 'Operational monitoring guidelines recommend exposure of 40 mosquitoes per LLIN and a total of 30 LLINs to be tested, resulting in a total of 1200 mosquitoes exposed. '

Methods: 'This approach used a lower number of mosquitoes per LLIN as recommended in the WHO guidelines for large scale (phase 3) LLIN candidate product evaluation where n=5 sections from n=30 unused LLINs are tested with n=20 mosquitoes each (i.e., n=100 mosquitos per LLIN) and resembled more closely the approaches recommended for testing LLINs under operational conditions.'

Both sentences are followed by the recommended reference (now reference number 24).

Comment 2. The reference section has some errors and needs fixing.

Response 2: We have reviewed and updated the reference section to ensure each reference complies with the formatting requirements.

Comment 3: Abstract: We hypothesise that decreased bioefficacy of LLINs is a major contributor to the malaria resurgence in PNG.

I would use a slightly stronger statement "These results suggest that decreased bioefficacy of LLINs is contributing to the malaria resurgence in PNG and increased scrutiny of LLIN quality is warranted."

Response 3: We modified the abstract slightly to incorporate this statement.

Comment 4. Discussion: This paper appears to have uncovered a potentially big problem in net quality since so many batches failed over such a large time frame. I would like to see the authors make a stronger recommendation in the discussion and call for quality assurance through operational monitoring of nets in all endemic countries to ensure that this is not a global problem.

Response 4: In order to address this comment we have added the following sentence to the discussion section.

'To investigate whether the observed problem is restricted to PNG or if other countries and LLIN products are also affected, we recommend that quality assurance through operational monitoring of LLINs in all malaria endemic countries is conducted and capacity of endemic countries to conduct the required studies is strengthened.'